# Effect of Black Garlic Consumption on Endothelial Function and Lipid Profile: A Before-and-After Study in Hypercholesterolemic and Non-Hypercholesterolemic Subjects

**DOI:** 10.3390/nu15143138

**Published:** 2023-07-14

**Authors:** Débora Villaño, Javier Marhuenda, Raúl Arcusa, José Manuel Moreno-Rojas, Begoña Cerdá, Gema Pereira-Caro, Pilar Zafrilla

**Affiliations:** 1Faculty of Health Sciences, Universidad Católica de San Antonio, 30107 Murcia, Spain or debora.villano@unavarra.es (D.V.); jmarhuenda@ucam.edu (J.M.); bcerda@ucam.edu (B.C.); mpzafrilla@ucam.edu (P.Z.); 2Department of Agroindustry and Food Quality, Andalusian Institute of Agricultural and Fisheries Research and Training (IFAPA), Alameda del Obispo, Avda. Menéndez-Pidal, 14004 Córdoba, Spain; josem.moreno.rojas@juntadeandalucia.es; 3Foods for Health Group, Instituto Maimónides de Investigación Biomédica de Córdoba (IMIBIC), 14004 Córdoba, Spain

**Keywords:** cardiometabolic diseases (CMDs), nutrition strategies, black garlic, organosulfur compounds, endothelial adhesion molecules

## Abstract

Background: Black garlic is obtained from raw garlic (*Allium sativum* L.), by a fermentation process, under humidity and heat treatment, showing a high concentration of organosulfur compounds, which have been related to benefits in the prevention or delay of cardiovascular diseases (CVDs). The objective of the research was to evaluate whether long-term consumption of black garlic improves endothelial function and lipid profile in subjects with hypercholesterolemia. Methods: Single center, controlled clinical trial with two branches: Hypercholesterolemia vs. Healthy condition. Sixty-two subjects of both sexes were distributed in two groups, the hypercholesterolemia group (*n* = 31) (total cholesterol (TC) range 200–300 mg/dL and low-density lipoprotein (LDL)-cholesterol range 135–175 mg/dL) and the healthy group (*n* = 31). The intervention consisted of the ingestion of 4 cloves of black garlic (12 g) daily for 12 weeks. Results: significant increases in Apolipoprotein (Apo)A1 occurred in both groups: Hypercholesterolemia (Δ 11.8 mg/dL *p* < 0.001) vs Healthy (Δ 11.1 mg/dL *p* < 0.001). Besides, significant reductions for endothelial adhesion molecules monocyte chemoattractant protein-1 (MCP-1) (Δ −121.5 pg/mL *p* = 0.007 vs. Δ −56.3 pg/mL *p* = 0.015), intracellular adhesion molecule-1 (ICAM-1) (Δ −39.3 ng/mL *p* < 0.001 vs. Δ 63.5 ng/mL *p* < 0.001), and vascular cyto-adhesion molecule-1 (VCAM-1) (Δ −144.4 ng/mL *p* < 0.001 vs. Δ −83.4 ng/mL *p* = 0.061) were observed, for hypercholesterolemic and healthy subjects, respectively. Conclusions: These data show that black garlic consumption could improve some parameters related to endothelial function and lipid profile, which may have a favorable impact on the risk of CVDs, although more long-term studies are necessary to confirm.

## 1. Introduction

Garlic (*Allium sativum* L.) is a plant food studied for its beneficial effects on the prevention and delay of the onset of certain diseases [1,2]. This is due to its composition of nutrients and bioactive compounds. In general, plants of the genus *Allium* are rich in vitamins, minerals, fiber, essential amino acids, phenolic compounds, and organosulfur compounds (OSCs) [3]. OSCs are responsible for their organoleptic characteristics (odor and taste) and within this group, the major and more extensively studied is allicin (diallyl thiosulfinate). Allicin is very unstable and transforms extremely rapidly into other compounds (diallyl mono-, di-, tri-sulfides, ajoene, dithiin) through different mechanisms [1,4]. Despite being beneficial to health, the consumption of garlic has decreased significantly probably because of its pungent smell and taste and it can cause gastrointestinal discomfort in some populations. For this reason, raw garlic is being used in different alternative forms: garlic nutraceuticals, smoked garlic, garlic oil, and black garlic [5,6,7].

Black garlic is obtained from raw garlic bulbs (*Allium sativum* L.) by a fermentation process at high temperatures (60–90 °C) and a humidity between 70–90%, with an incubation period of 10–80 days [8]. Humidity, fermentation time, and temperature vary depending on the manufacturer and affect the composition of bioactive substances [2,9]. These conditions change organoleptic properties turning garlic into black color, sweet, and completely odorless. The unpleasant odor of fresh garlic is due to allicin, which is converted into other compounds as diallyl sulfide derivatives [2]. Besides, the elevated temperature required to produce black garlic is responsible for the changes in bioactive compounds, showing an increase in S-alkyl-cysteine compounds (SAC), coumaric acid, total phenols, and flavonoids [10]. It should be taken into account that if the temperature is high for a prolonged period of time, a decrease in these bioactive compounds could occur [10,11].

In a comparative analysis of fresh and black garlic after simulated in vitro digestion [12], the bioaccessibility of polyphenols and OSCs, including SAC and γ-glutamyl-S-alkyl-cysteine derivatives (GSAk), was affected during the different digestion phases (oral, gastric and intestinal). For polyphenols, the mean bioaccessibility in fresh garlic (58.6%) was slightly higher than in black garlic (47.2%), while for OSCs it was clearly higher for black garlic (55.3%) being (62.3% SACs vs. 42.7% GSAk) than for fresh garlic (15.3%) being (12% SACs vs. 26.3% GASk). The main polyphenol compounds found after the whole digestive process in fresh garlic were vanillic acid (69.4%) and ferulic acid (51.5%), while in black garlic it was caffeic acid (64.8%). For OSCs, it is worth mentioning that aliin presented a 7-fold more bioaccessibility for black garlic (76.6%) vs. fresh garlic (5.3%) after the digestion process Therefore, the fermentation process of fresh garlic to obtain black garlic has positive effects by increasing the bioaccessibility of OSCs.

OSCs contribute to the beneficial properties of garlic for their anti-inflammatory, antioxidant, and antidiabetic activities [2]. It has been observed that garlic can reduce oxidative stress by activating antioxidant enzymes, apoptosis, inflammation, mitochondrial dysfunction, and autophagy that occur as a consequence of cardiac injury induced by antineoplasic drugs [13]. In vitro studies of aged black garlic extracts have shown the inhibition of thyroid cancer cell lines and lung cancer Lewis cells [14,15]. Vlachojannis et al. showed that the consumption of black garlic improves different cardiovascular risk factors such as blood pressure, cholesterol levels, and slows down the atherogenic process [16]. Consumption of black garlic has shown cardioprotective effects by reducing atherogenic markers [17]. However, in vivo studies on the effect of black garlic on a population at cardiovascular risk are scarce, compared to raw garlic. Our hypothesis is that black garlic is able to attenuate the disbalance of lipid profile and improve endothelial function in the condition of hypercholesterolemia. Thus, the aim of the present study was to evaluate the effect of black garlic consumption on markers of cardiovascular function in hypercholesterolemic adult populations, compared to healthy subjects.

## 2. Materials and Methods

### 2.1. Clinical Trial Design

The clinical trial consisted of a single-center, chronic, controlled study with two parallel branches according to health condition (hypercholesterolemic and healthy) (Figure 1). The intervention had a duration of 12 weeks for each group.

During the intervention, the volunteers came to the laboratory two times (at the beginning and at the end of the nutritional intervention). During the visits, a blood sampling was performed, blood pressure and anthropometric parameters were analyzed, and two questionnaires were carried out, one on physical activity and the other on a 24-h (3-day) dietary survey.

### 2.2. Participants

A total of 62 subjects of both sexes, aged between 40 and 64 years, were included. In order to participate in the study, they had to meet all the following selection criteria: age between 40 and 65 years; Body Mass Index (BMI) between 25–30 kg/m^2^; healthy group (TC levels < 200 mg/dL; LDL-cholesterol levels < 135 mg/dL); hypercholesterolemia group (TC levels between 200–300 mg/dL; LDL-cholesterol levels between 135–175 mg/dL). Exclusion criteria were: not being under pharmacological treatment; not consuming hormones; not consuming dietary supplements; not smokers; not being pregnant; not vegetarians; not being under dietary treatment. All participants signed the informed consent.

In order to participate in the study, all subjects followed some hygienic-dietary recommendations that were explained by the investigators prior to the beginning of the study: not to start or modify any hormonal treatment during the study if it was not duly justified; not to significantly modify dietary and physical activity habits; not to take or follow any treatment that could affect the study parameters. Lifestyle questionnaires (GPAQ or Global Physical Activity Questionnaire) [18] to determine the level of sedentary lifestyle or activity and total qualitative and quantitative intake of food ingested with a 24-h recall dietary questionnaire (24-h dietary survey), including two weekdays and one weekend day were collected, to control that there were no changes on dietary and lifestyle habits.

To carry out the blood extractions, the following instructions were set: subjects had to come after fasting for a minimum of 12 h, only allowing the intake of water in the 3 h prior to the extraction and they could not perform moderate-intense exercise at least 24 h prior to blood collection.

### 2.3. Test Product

The product under study consisted of black garlic cloves. The product was supplied by the Institute for Agricultural and Fisheries Research and Training (IFAPA, Córdoba, Spain) and the local supplier (La Abuela Carmen, Córdoba, Spain).

Each participant had to ingest 4 cloves per day (preferably at breakfast) for 12 weeks.

The nutritional information of the product (100 g) is: energy value 218 kcal; carbohydrates 45 g, of which sugars 30 g; dietary fiber 3 g; protein 9.5 g; fat 0 g and salt 0.025 g.

### 2.4. Biochemical Variables

All variables were analyzed in the study population, at the beginning and at the end of the 12 weeks of uninterrupted consumption of the product.

A venous blood extraction was performed from one of the forearm veins close to the elbow flexure after the subject had fasted for 12 h. Samples were collected in Vacutainer^®^ tubes and were centrifuged at 4500 rpm for 5 min at 4 °C, subsequently aliquots were stored frozen at −80 °C until analysis.

Regarding the variables analyzed, the following were determined:(a)Lipid profile: apolipoproteins (Apo) A1 and B were analyzed in serum by standardized methods in an automated immunoassay system Unicel Dxl 800 Access. TC, triglycerides, and high-density lipoprotein (HDL)-cholesterol was measured in serum by enzymatic methods using the cholesterol oxidase-peroxidase technique in a BA 400 BioSystems analyzer. LDL-cholesterol was calculated using the Friedewal formula [19], and the atherogenic index (TC/HDL-cholesterol,) and the ApoA1/ApoB ratio were calculated.(b)Endothelial function biomarkers: MCP-1, ICAM-1, VCAM-1, E-Selectin and P-Selectin were measured in serum using commercial enzyme immunoassay kits (ELISA, R&D Systems, Abingdon, UK) and Nitric Oxide (NO) was determined through a colorimetric method, following the protocols [20,21].

Safety parameters such as leukocyte series, blood count, thyroid hormones, and transaminases were also evaluated.

### 2.5. Blood Pressure

Systolic and diastolic blood pressure were monitored using a digital sphygmomanometer, to determine the impact of black garlic consumption. Measurements were performed in triplicate, after a minimum rest of 15 min, leaving 5 min in between.

### 2.6. Body Composition

Bioimpedance was carried out to evaluate the changes in body composition of the individuals throughout the study, based on the electrical conduction properties of the biological tissues. This measurement was carried out under fasting conditions, at the beginning (baseline visit) and at the end of the study. For this purpose, a TANITABCV-454 Body Composition Analyzer was used.

### 2.7. Statistical Analysis

The size of the study population was estimated taking total blood cholesterol level as the main variable. Considering a statistical power of 80%, a statistical significance level of 5% (one tail), a standard deviation of 6.5, aiming to estimate a difference of 2.5, the number of volunteers required is 25, rising to 31 in anticipation of dropouts for each group (Healthy and Hypercholesterolemic). Descriptive analysis of all the variables under study (mean and standard deviation), on baseline conditions and after the treatment were obtained on each group.

Comparative inter-group analysis: A comparison of values of variables at the beginning of the intervention was performed between the healthy and hypercholesterolemic groups, to check for initial differences, by means of a *t*-Student test. Inter-group comparison of the effect of treatment was performed with the *t*-Student test on the changes produced in each variable (final value-initial value).

Comparative intra-group analysis: To observe the changes produced in the variables over time, a paired *t*-Student test was performed with an intrasubject factor (time: baseline and final).

For the statistical analysis, the level of significance used was 0.05, and the SPSS program (version 19.0; SPSS, Inc., Chicago, IL, USA, IBM Company) was used.

### 2.8. Ethical Approval

The protocol was approved by the Institutional Review Committee of the Catholic University San Antonio of Murcia (UCAM) (date: 26 June 2020; code CE062008). This study was carried out following the Standards of Good Clinical Practice and was conducted according to the Declaration of Helsinki. The trial was registered at www.clinicaltrials.gov (accessed on 18 October 2021) (identifier NCT05082350). The study was carried out in the Pharmacy Department of the Faculty of Health Sciences of UCAM. Current European legislation on the protection of personal data (Regulation (EU)2016/679) was complied with.

## 3. Results

### 3.1. Study Population

A total of 62 subjects of both sexes (31 men and 31 women) participated in the intervention, divided into 31 subjects with hypercholesterolemia and 31 healthy subjects as the control group. Figure 2 depicts the flow diagram. Table 1 shows that, except for TC, LDL-cholesterol and HDL-cholesterol values, both groups were similar for the rest of the variables.

### 3.2. Lipid Profile

Table 2 shows all the variables related to the lipid profile assessed during the intervention.

The consumption of black garlic increased significantly (*p <* 0.001) the plasma concentration of ApoA1 in both groups with a mean increase of 7.6% for the hypercholesterolemia group and 7.4% for the control group. No statistically significant differences were observed between groups at the end of the intervention. Regarding plasma ApoB levels, a significant increase of 6.9% (*p <* 0.05) was observed in the hypercholesterolemia group, with statistical differences with the healthy group at the end of the intervention (*p <* 0.001). Since the levels of these lipoproteins are closely related, the ApoB/ApoA1 ratio was calculated for each subject. There were significant differences in the baseline values, although no effects were observed with the consumption of black garlic in any of the studied groups. TC values were increased in both groups after the consumption of black garlic, which was significant in the healthy group (*p <* 0.05). Despite this, values were within the range considered non-pathological, below 200 mg/dL. At the end of the intervention, significant differences were observed between groups. Analyzing the different cholesterol fractions, LDL-cholesterol values remained stable without significant changes in the hypercholesterolemic group, while their values increased significantly (*p <* 0.05) in the healthy group. However, these increases were in the range considered as a low-risk level (LDL-cholesterol < 130 mg/dL). HDL-cholesterol values significantly increased in healthy subjects (*p <* 0.001) as well. No statistically significant differences were observed between groups. The atherogenic index was obtained by calculating the ratio of TC/HDL-cholesterol for each subject. For the present variable, statistically significant differences were observed at the end of the intervention, with lower values in the healthy vs. the hypercholesterolemia subjects. The healthy group always showed values considered at low risk (3.3–4.4), and the hypercholesterolemia group dropped to values considered the lower limit for medium risk (which is between 4.4–7.1). Regarding triglycerides, no significant changes were observed with garlic consumption, in any of the groups studied.

### 3.3. Endothelial Function

Table 3 shows all the variables related to the endothelial function assessed during the intervention. Comparing values at the beginning of the intervention, ICAM-1 was significantly different (*p* < 0.05).

MCP-1 levels significantly decreased (*p <* 0.05) in both groups after black garlic consumption, decreasing by 13.9% and 7.2% in the hypercholesterolemia and healthy groups, respectively. At the end of the intervention, no statistically significant differences were observed between groups.

VCAM-1 levels were reduced in both groups, being only significant in the hypercholesterolemia group (*p <* 0.05) with a 12.5% decrease. No differences were observed between groups.

ICAM-1 levels were significantly reduced in both groups after the consumption of black garlic (*p <* 0.001), being 10.6% in the group with hypercholesterolemia and 15.7% in the healthy group. These reductions were significantly different comparing the groups at the end of the intervention (*p <* 0.05).

By contrast to the results observed for the previous variables, P-Selectin and E-Selectin levels significantly increased in both groups after the consumption of black garlic. For P-Selectin, increases of 8% (*p <* 0.002) and 17.4% (*p <* 0.001) were observed in the hypercholesterolemia and healthy group respectively, and 17.4% (*p <* 0.001) in the control group, with no significant differences between them. For E-selectin, increases of 7.9% (*p <* 0.05) were observed in hypercholesterolemia and 13.9% (*p <* 0.002) in the healthy group, being significantly different between the two groups (*p <* 0.014). Nitric oxide values did not change significantly in any of the groups studied.

### 3.4. Blood Pressure and Anthropometric Parameters

After black garlic consumption, there were no significant variations in blood pressure, neither in systolic (SDP) nor diastolic pressure (DBP) (Table 4). In terms of anthropometric values, no significant changes were observed in weight, BMI, fat mass, or fat-free mass (Table 4).

In addition, no significant changes were observed in blood counts, transaminases, and thyroid function and their values were within the normal ranges, hence it can be affirmed that the consumption of the product under investigation is safe.

## 4. Discussion

The present study evaluates the effect of black garlic consumption, in dietary portions, on the lipid profile and endothelial function in subjects with hypercholesterolemia, as well as in healthy individuals. The recommended daily dose for the therapeutic use of garlic is 2–4 g of crushed raw garlic or equivalent in alliin content [22].

With the manufacturing conditions of black garlic, favoring Maillard reactions between proteins and sugars, the chemical composition of raw garlic changes, including the type and concentration of bioactives, which has a major impact on its biological properties [23].

### 4.1. Effects of Black Garlic on Lipid Profile

The consumption of black garlic significantly increased the levels of both HDL-cholesterol and LDL-cholesterol in healthy subjects, while the hypercholesterolemia group did not vary significantly. LDL-cholesterol particles are associated with a worse progression of CVD while HDL-cholesterol particles remove cholesterol from the tissues and high levels in blood are considered beneficial for vascular function. To better understand the relationship between the changes of both lipoproteins, the atherogenic index (TC/HDL-cholesterol) was calculated for each subject, showing a significant reduction in the healthy group after back garlic treatment. Therefore, the changes in TC, HDL-cholesterol, and LDL-cholesterol levels observed, when assessed individually, indicate that the consumption of black garlic could improve the lipoprotein profile in this group.

The consumption of garlic acid increased levels of Apo A1 and Apo B lipoproteins. The ApoA1 protein is part of the HDL-cholesterol structure, so its increase is positively associated with the presence of these lipoproteins. On the other hand, Apo B is part of the structure of LDL-cholesterol and serum ApoB levels reflect the total number of these lipoproteins, which excess has negative consequences for health [24,25]. It has been stated that the ratio of ApoB to Apo A1 is a predictive measure of atherogenicity [26]. In our study, the ApoB/ApoA1 ratio was calculated for each individual and the decrease observed was not significant in any group. In contrast, other studies have shown significant decreases in this ratio in subjects with mild hypercholesterolemia after consumption of 6 g/day of AGE for 12 weeks [17].

Human studies on the effects of garlic on blood lipid profiles show contrasting results. Some of them have demonstrated that the administration of aged black garlic or garlic tablets has positive effects on dyslipidemia. Significant reductions in TC and LDL-cholesterol were observed in obese patients taking garlic supplements (800 mg/day) for 3 months [27]. Sobenin et al. [28] showed an anti-hypercholesterolemic effect after 12 months of treatment with 300 mg/day of garlic powder, with reductions in LDL-cholesterol in patients with coronary artery disease. Similarly, reductions in TC and LDL-cholesterol were observed in type 2 diabetes mellitus (T2DM) patients after 12-week consumption of 600 mg/day [29]. In contrast, other authors have reported no significant effects on lipid profile in overweight smokers at a dose of 2.1 g/day over a period of 3 months [30], mild hypercholesterolemic subjects taking 1.4 g/day garlic powder for 6 months [31], nor in type 2 diabetic patients at doses of 1.2 g/day for 4 weeks [32].

Current meta-analysis studies of RCT of the effects of garlic on CVD showed that TC and LDL-cholesterol significantly diminished with lower doses (<1.1 g equivalents of garlic extract) and longer duration (>90 days) [33]. In contrast, another meta-analysis has suggested that the evidence of efficacy is low for doses of 300–600 mg per day of garlic powder, in populations with TC levels > 6.0 mmol/L (231 mg/dL). Statistically significant reductions in fractions of lipid profiles were reported only in 2 of the 15 trials that studied garlic in the management of dyslipidemia [34]. In those cases, time-released garlic powder tablets were used, at doses of 600 mg/day, with interventions lasting from 12 weeks up to 12 months [28,35]. One of the main limitations pointed out in this meta-analysis is the lack of accurate product characterization, therefore the outcomes from clinical data cannot be transferred to other fermented or aged garlic products [16].

The discrepancies observed in the literature can be explained by differences in the study design, dosage, and duration of supplementation, as well as metabolic status of individuals at the beginning. The type of garlic supplement used (raw, powder, oil, aged garlic extract) may yield different results because of differences in the content of bioactives and their metabolites. In this sense, standardized products with a high content of organosulfur compounds are commercially available and have been studied, such as ABGO10+^®^ (black garlic extract concentrated in SAC and melanoidins), Kyolic^®^ (aged garlic extract rich in alliin), among others.

The phytochemical composition of the black garlic that has been used in the present study was previously investigated. p-Coumaric acid, caffeic acid, and gallic acid are the main phenolic acids present in black garlic, representing 99% of the total phenolic content. OSCs are present, mainly GSAk derivatives (31.1% of total OSCs), and SAC derivatives (64.1% of total OSCs) [12]. Four cloves (12 g) of black garlic provided an amount of 64.8 mg of total organosulfur compounds, of which 14.4 mg are total SACs derivatives and 11.98 μg allicin, and 684 mg of phenolic compounds. In vitro digestion and subsequent colonic fermentation have shown that these compounds are metabolizable and absorbable and could therefore be responsible for the observed effects [36]. Human bioavailability studies are necessary to elucidate the responsibility of the absorbed compounds and their metabolites for the observed physiological response.

### 4.2. Endothelial Function

Specific adhesion molecules such as VCAM-1, ICAM-1, and E-selectin are expressed on the surface of vascular endothelium and mediate in monocyte recruitment to the vessel wall. Plasma levels of these adhesion molecules help to predict the onset and prognosis of chronic diseases with an endothelial dysfunction component. In this sense, a longitudinal study on type 1 diabetes showed that higher levels of VCAM-1 (but not E-selectin or C-reactive protein) at baseline and through the 20-year follow-up period predicted the prevalence and incidence of hypertension [37]. In our study, we observed significant decreases in levels of VCAM-1 and ICAM-1 in both healthy and hypercholesterolemic groups, whilst selectin levels significantly increased. Previous studies showed that aged garlic increased flow-mediated endothelium-dependent dilation (FMD) in men with coronary artery disease (CAD) after a 2-wk treatment, with no significant changes in VCAM-1 levels and lipid profile. The intervention consisted of capsules containing 600 mg of aged garlic extract, with a daily dose of 2.4 g [38]. In vitro studies have shown that garlic components suppress VCAM-1 and ICAM-1 expression in HCAEC endothelial cell cultures, decreasing monocyte adherence [39], by affecting signaling pathways [40]. Similarly, garlic extracts effectively suppressed the expression of VCAM-1, the activation of NF-κβ transcription factor, and monocyte adhesion [41]. No data have been reported up to date on changes in selectins with black garlic treatment. Moreover, in our study black garlic consumption decreased the levels of MCP-1 in both populations studied. MCP-1 is a chemokine released by the endothelium that regulates monocytes and leukocytes migration and infiltration. It has been associated with the development of atheroma plaques and different pathologies with inflammatory components. Its decrease in black garlic consumption could contribute to reducing the inflammatory response and its progression.

### 4.3. Blood Pressure

Regarding blood pressure, no significant changes were observed in either group. These results are in contrast with the literature, that has shown antihypertensive effects using extracts of black garlic or aged black garlic [16,42]. In this sense, in a randomized crossover clinical trial, an extract of 250 mg of aged black garlic containing 1.25 mg of SAC was administered to a moderately hypercholesterolemic population for 6 weeks, showing a 4.82 mm Hg reduction in DBP compared to placebo, when baseline DBP was greater than 75 mm Hg [43]. Although, as shown in Table 1, the baseline DBP values for our subjects were also above 75 mm Hg, no such effect of garlic consumption was observed. An antihypertensive effect was also observed after 12 weeks of consumption of an aged black garlic extract (960 mg/day with a content of 2.4 mg SAC): SBP was significantly reduced by 10.42 ± 4.3 mm Hg only in patients with SBP > 140 mm Hg at baseline. These baseline values are higher than those of our subjects (Table 1) [44]. Similarly, an intervention in which 1.2 mg SAC/day was administered for 12 weeks, showed a reduction in SBP and DBP when baseline values were higher than 150 mm Hg and 93 mm Hg, respectively [45]. These findings are in contrast with ours, possibly because our population had lower values for both SBP and DBP. In the clinical trial of Valls et al. [43] no significant changes were observed in anthropometric or vasodilator measurements, similar to our study.

Ried et al. [46] showed that after 12 weeks of intake of an aged black garlic extract containing 1.2 mg SAC, mean arterial blood pressure was significantly reduced by 5.0 ± 2.1 mm Hg, the reduction being 11.5 ± 1.9 mm Hg for SBP and 6.3 ± 1.1 mm Hg for DBP. A meta-analysis involving 12 trials and 553 adults with high blood pressure suggests that different garlic supplements significantly reduce SBP by an average of 8.3 ± mm Hg and DBP by 5.5 ± 1.9 mm Hg [42]. However, these trials were conducted in heterogeneous groups of patients aged over 60 years, hypertensive, obese or overweight, uncontrolled, some of them with comorbidities and receiving frequent pharmacological treatment, and the majority had moderate to very high CVD. One of the limitations of our research, that could explain the null effects observed in blood pressure, is that our subjects did not suffer from hypertension. It is possible that if the study had been carried out in people with hypertension, significant changes in this variable would have been observed.

## 5. Conclusions

The present study indicates possible beneficial effects of daily intake of black garlic. Consumption of black garlic improved the ratio of TC/HDL-cholesterol in healthy subjects and there was a tendency to decrease in the atherogenic index of apolipoproteins. Decreased levels observed of endothelial adhesion molecules, VCAM-1 and ICAM-1, may indicate a better prognosis for complications of hypercholesterolemia. However, long-term studies are needed to confirm the relevance of these findings, examining the effect in a cohort at increased cardiovascular risk. It is also necessary to evaluate the bioavailability of low molecular weight bioactive compounds from black garlic, in order to correlate their presence with the observed biological effects.

## Figures and Tables

**Figure 1 nutrients-15-03138-f001:**
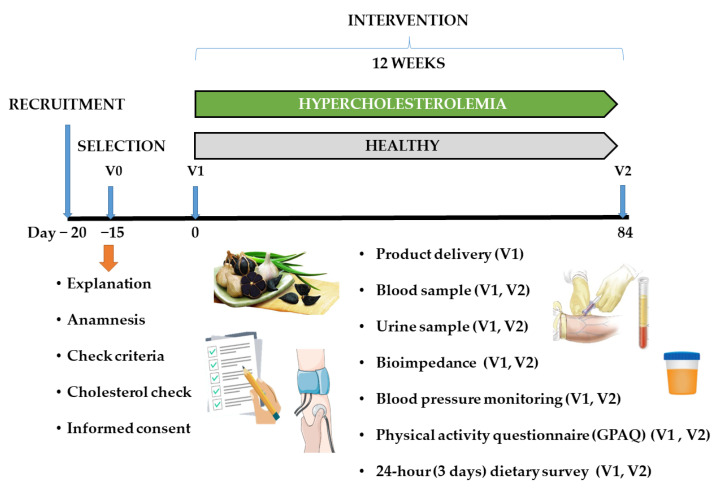
Graphical representation of the clinical trial.

**Figure 2 nutrients-15-03138-f002:**
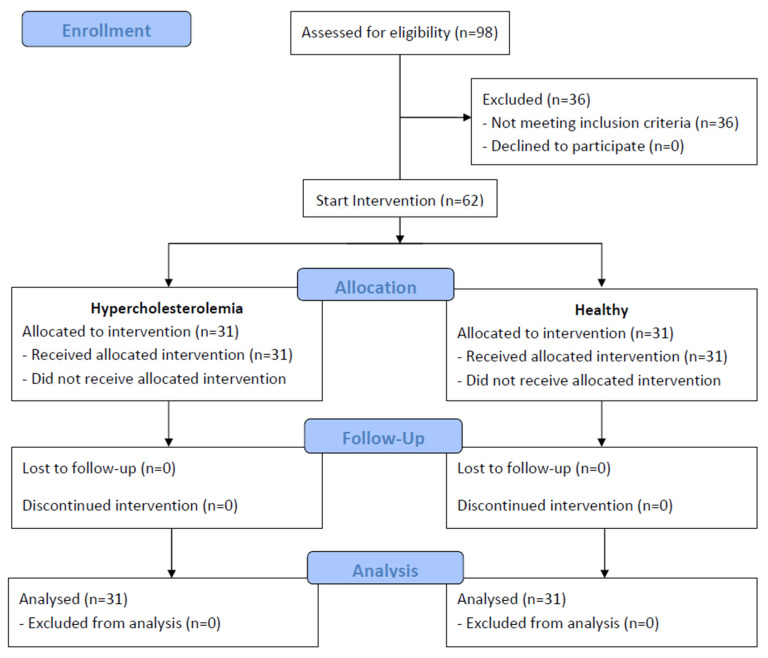
Flow diagram of the study.

**Table 1 nutrients-15-03138-t001:** Baseline characteristics of the subjects (mean and standard deviation).

	Hypercholesterolemia	Healthy	*p*-Value
*n*	36	36	1
Age	47.5 ± 6.7	46.6 ± 7.5	0.595
Weight (kg)	82.5 ± 10.8	82.2 ± 12	0.938
Height (m)	1.7 ± 7.9	1.7 ± 9.6	0.219
BMI (kg/m^2^)	27.1 ± 2.5	27.8 ± 2.2	0.289
Fat Mass %	27.3 ± 6.9	29.1 ± 7.6	0.325
Free Fat Mass (kg)	57.1 ± 9.8	55.4 ± 10.6	0.513
SBP (mm Hg)	123 ± 14.1	125.7 ± 14.3	0.465
DBP (mm Hg)	82.1 ± 8.7	84.3 ± 10.2	0.377
Total cholesterol (mg/dL)	238.1 ± 29.2	177.3 ± 16.9	<0.001 ^†^
LDL-cholesterol (mg/dL)	160.3 ± 24.8	108.6 ± 17.9	<0.001 ^†^
HDL-cholesterol (mg/dL)	55.5 ± 13.2	48.2 ± 11.8	<0.025 ^†^
Triglycerides (mg/dL)	97.3 ± 31.7	96.2 ± 52.7	0.919

^†^ Significant differences between groups at baseline (*p* < 0.05).

**Table 2 nutrients-15-03138-t002:** Changes on lipid profile biomarkers after black garlic consumption (mean and standard deviation).

Biomarker	Group	Baseline	Final	Inter-Group *p*-Value Baseline	Intra-Group *p*-Value Time	Inter-Group *p*-Value Final
ApoA1 (mg/dL)	Hypercholesterolemia	155.7 ± 15.3	167.5 ± 15.4	0.218	<0.001 *	0.172
	Healthy	149.7 ± 21.7	160.8 ± 19.4		<0.001 *	
ApoB (mg/dL)	Hypercholesterolemia	111.6 ± 22	119.3 ± 21.6	<0.001 ^†^	0.029 *	<0.001 ^#^
	Healthy	82.3 ± 11.9	85.8 ± 13		0.119	
ApoB/ApoA1 (mg/dL)	Hypercholesterolemia	0.73 ± 0.2	0.72 ± 0.2	<0.001 ^†^	0.870	0.596
	Healthy	0.56 ± 0.1	0.54 ± 0.1		0.301	
Total cholesterol (mg/dL)	Hypercholesterolemia	238.1 ± 29.2	243.3 ± 31	<0.001 ^†^	0.325	0.03 ^#^
	Healthy	177.32 ± 16.9	191.7 ± 18.9		<0.001 *	
LDL-cholesterol (mg/dL)	Hypercholesterolemia	160.3 ± 24.75	161.3 ± 22.3	<0.001 ^†^	0.810	0.325
	Healthy	108.6 ± 17.9	118.3 ± 17.4		0.006 *	
HDL-cholesterol (mg/dL)	Hypercholesterolemia	55.5 ± 13.2	57.1 ± 10.1	0.025 ^†^	0.284	0.097
	Healthy	48.2 ± 11.8	56 ± 11.2		<0.001 *	
Triglycerides (mg/dL)	Hypercholesterolemia	97.3 ± 31.7	100.5 ± 40.4	0.919	0.415	0.408
	Healthy	96.1 ± 52.7	86.9 ± 38.2		0.171	
Atherogenic Index (T. cholesterol/HDL)	Hypercholesterolemia	4.52 ± 1.2	4.4 ± 1	0.015	0.495	0.044 ^#^
	Healthy	3.86 ± 0.9	3.54 ± 0.7		0.022	

^†^ Significant differences between groups at baseline (*p* < 0.05). *** Significant intra-group differences during the intervention (*p* < 0.05). ^#^ Significant differences between groups at the end of the intervention (*p* < 0.05).

**Table 3 nutrients-15-03138-t003:** Changes in endothelial function biomarkers after black garlic consumption (mean and standard deviation).

Biomarker	Group	Baseline	Final	Inter-Group *p*-Value Baseline	Intra-Group *p*-Value Time	Inter-Group *p*-Value Final
MCP-1 (pg/mL)	Hypercholesterolemia	875.4 ± 224.5	753.9 ± 240	0.120	0.007 *	0.225
	Healthy	779.1 ± 255.1	722.8 ± 242.2		0.015 *	
VCAM-1 (ng/mL)	Hypercholesterolemia	1157.3 ± 217	1012.9 ± 202.9	0.655	<0.001 *	0.433
	Healthy	1124.4 ± 344.8	1041 ± 301.5		0.061	
ICAM-1 (ng/mL)	Hypercholesterolemia	370.2 ± 53.2	330.9 ± 53	0.034	<0.001 *	0.038 ^#^
	Healthy	403.7 ± 67.9	340.2 ± 51.3		<0.001 *	
P-Selectin (ng/mL)	Hypercholesterolemia	161.9 ± 29.9	174.8 ± 31.9	0.244	0.002 *	0.345
	Healthy	152.6 ± 32.5	179.1 ± 39.2		<0.001 *	
E-Selectin (ng/mL)	Hypercholesterolemia	52.9 ±16.8	57.1± 16.7	0.146	0.012 *	0.014 ^#^
	Healthy	59.5 ± 17.5	67.8 ± 18.7		0.002 *	
Nitric Oxide (µmol/L)	Hypercholesterolemia	4.7 ± 3.3	5.8 ± 5.4	0.470	0.297	0.839
	Healthy	3.9 ± 3.2	4.7 ± 3.8		0.360	

* Significant intra-group differences during the intervention (*p* < 0.05). ^#^ Significant differences between groups at the end of the intervention (*p* < 0.05).

**Table 4 nutrients-15-03138-t004:** Changes in blood pressure and anthropometric parameters after black garlic consumption (mean and standard deviation).

Biomarker	Group	Basal	Final	Inter-Group *p*-Value Baseline	Intra-Group *p*-Value Time	Inter-Group *p*-Value Final
SBP (mmHg)	Hypercholesterolemia	123.00 ± 14.11	127.61 ± 17.02	0.465	0.153	0.577
	Healthy	125.67 ± 14.34	127.22 ± 12.68		0.396	
DBP (mmHg)	Hypercholesterolemia	82.13 ± 8.68	82.41 ± 10.36	0.377	0.840	0.515
	Healthy	84.29 ± 10.16	85.35 ± 9.05		0.446	
Weight (kg)	Hypercholesterolemia	82.45 ± 10.83	81.97 ± 10.13	0.938	0.946	0.425
	Healthy	82.22 ± 12.04	82.86 ± 12.47		0.217	
BMI (kg/m^2^)	Hypercholesterolemia	27.13 ± 2.49	27.02 ± 2.44	0.289	0.905	0.428
	Healthy	27.77 ± 2.19	27.99 ± 2.37		0.201	
Fat mass (%)	Hypercholesterolemia	27.27 ± 6.98	28.02 ± 6.73	0.325	0.661	0.361
	Healthy	29.12 ± 7.57	29.88 ± 7.59		0.053	
Free fat mass (kg)	Hypercholesterolemia	57.10 ± 9.80	56.57 ± 9.59	0.513	0.583	0.630
	Healthy	55.38 ± 10.62	54.96 ± 10.48		0.396	

## Data Availability

The data are contained within the article.

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
