# Peer review of "Effect of Black Garlic Consumption on Endothelial Function and Lipid Profile: A Before-and-After Study in Hypercholesterolemic and Non-Hypercholesterolemic Subjects"

_nutrients, 2023, doi:10.3390/nu15143138_

Round 1

Reviewer 1 Report (Previous Reviewer 2)

The revised version of the manuscript entitled "Effect of black garlic consumption on endothelial function and lipid profile" [Effect of black garlic consumption on endothelial function and lipid profile: a before-and-after study in hypercholesterolemic and non-hypercholesterolemic subjects] demonstrates commendable efforts in addressing the concerns raised in the previous review. I have carefully considered the changes made and compared them to the previous version, and I am confident in stating that the revised manuscript is significantly better than its predecessor. Specifically, the authors have successfully clarified several aspects of the manuscript. The writing has become more concise and clear, which has greatly improved the overall readability. The logical flow and organization of the paper have been strengthened, resulting in a more coherent and engaging narrative.

Author Response

Reviewer 1

The revised version of the manuscript entitled "Effect of black garlic consumption on endothelial function and lipid profile" [Effect of black garlic consumption on endothelial function and lipid profile: a before-and-after study in hypercholesterolemic and non-hypercholesterolemic subjects] demonstrates commendable efforts in addressing the concerns raised in the previous review. I have carefully considered the changes made and compared them to the previous version, and I am confident in stating that the revised manuscript is significantly better than its predecessor. Specifically, the authors have successfully clarified several aspects of the manuscript. The writing has become more concise and clear, which has greatly improved the overall readability. The logical flow and organization of the paper have been strengthened, resulting in a more coherent and engaging narrative.

We deeply appreciate your comments.

Reviewer 2 Report (New Reviewer)

Based on the high concentration of organic sulfur in black garlic obtained by heat treatment under suitable humidity conditions, and the known benefits of organic sulfur in preventing or delaying cardiovascular disease, a 12-week dietary intervention was conducted on 62 subjects to evaluate whether black garlic can improve the health of patients with hypercholesterolemia. The results show that long-term consumption of black garlic can improve some parameters of endothelial function and blood lipids, which may have a beneficial effect on cardiovascular disease. The logic of the experimental analysis in this paper is relatively complete, and it can be accepted after appropriate improvement. If the paper wants to further improve the quality, it can be considered to organize dietary intervention for hypertensive patients. If conditions permit, the group of subjects for dietary intervention with black garlic can be further expanded in terms of age and disease types, which will make the literature have higher reference value. 

Concerns: It is also mentioned in the article that the conditions under which black garlic is produced by different manufacturers may be different. Is the organic sulfur content of black garlic produced under different conditions also different? If the organic sulfur content of black garlic under different production conditions is quite different, it can be considered to use several black garlic under different production conditions for analysis and experiment, so as to have greater reference significance. In addition, the paper also mentioned that due to the decrease of diastolic and systolic blood pressure in the population in this trial, there was no significant antihypertensive effect in the subjects, if the authors wish

Minors: The format of the diagram in the text needs to be uniform to make it look good.

Author Response

Reviewer 2

Based on the high concentration of organic sulfur in black garlic obtained by heat treatment under suitable humidity conditions, and the known benefits of organic sulfur in preventing or delaying cardiovascular disease, a 12-week dietary intervention was conducted on 62 subjects to evaluate whether black garlic can improve the health of patients with hypercholesterolemia. The results show that long-term consumption of black garlic can improve some parameters of endothelial function and blood lipids, which may have a beneficial effect on cardiovascular disease. The logic of the experimental analysis in this paper is relatively complete, and it can be accepted after appropriate improvement. If the paper wants to further improve the quality, it can be considered to organize dietary intervention for hypertensive patients. If conditions permit, the group of subjects for dietary intervention with black garlic can be further expanded in terms of age and disease types, which will make the literature have higher reference value. 

We deeply appreciate your comments. Regarding your suggestions to improve the quality of the manuscript, since it is not possible for us to carry them out in the present research, we will take them under consideration for the future studies. We agree that the future research line needs to increase the ranges of age and diseases related to cardiovascular conditions, for which we are searching for project funding sources.

Concerns: It is also mentioned in the article that the conditions under which black garlic is produced by different manufacturers may be different. Is the organic sulfur content of black garlic produced under different conditions also different? If the organic sulfur content of black garlic under different production conditions is quite different, it can be considered to use several black garlic under different production conditions for analysis and experiment, so as to have greater reference significance. In addition, the paper also mentioned that due to the decrease of diastolic and systolic blood pressure in the population in this trial, there was no significant antihypertensive effect in the subjects, if the authors wish

Certainly, there is a fermentation process with temperatures ranging from 60 to 90°C and humidities of 70-90%, as well as days of incubation, so each manufacturer can use these ranges depending on their methodology and working equipment. In this sense, black garlic was provided by a local manufacturer that has optimized its conditions of manufacturing. The technology may have an impact on the content of organosulfur compounds, but its evaluation was not in the scope of our study. Instead, we made a complete characterization of the phytochemical composition of our black garlic, that has been already published, to know exactly the components tested in the human study. Nevertheless, it could be interesting to carry out future interventions with parallel branches with different commercial types of black garlic, as well as to perform an assessment of the impact of the different variables of the fermentation process in the content of the particular organosulfur compounds.

In accordance with the advice of previous reviewers, it was decided not to expand on variables that did not significantly change, such as blood pressure.

Minors: The format of the diagram in the text needs to be uniform to make it look good.

We do not fully understand what you are referring to. We have checked and improved Figure 2 of the flow diagram.

Round 2

Reviewer 2 Report (New Reviewer)

Accept

This manuscript is a resubmission of an earlier submission. The following is a list of the peer review reports and author responses from that submission.

Round 1

Reviewer 1 Report

Thank you very much for the opportunity to review the article entitled “Effect of black garlic consumption on endothelial function and lipid profile”. In this paper, the author attempted to explore the effects of black garlic intake on cardiovascular risk factors and lipid profile through clinical trials. This is a study of practical significance, but there are defects in experimental design and other aspects, leading to the lack of credibility of experimental data.

I have some questions and suggestions for the authors.

1. The study lacked a negative control group, and a group of hypercholesterolemia participants should be set up without black garlic intake as a negative control.

2. in line 157, the authors claim that there were no differences in diet and lifestyle between the control and intervention groups. However, table 4 reported that in the control group, levels of MCP-1, VCAM-1, and ICAM-1 were significantly decreased after 12 weeks. These results indicated that lifestyles such diets and physical activities may improve the Endothelial function biomarkers in both control and intervention groups. It was not clear whether the results in the intervention group were due to lifestyle change or black garlic consumption. Moreover, increased level of some biomarkers such as P-Selectin as E-Selectin in the intervention group casts doubt on the beneficial effects of black garlic consumption.

3. Table 3 reported increased levels of apoA1, total cholesterol, and LDL-cholesterol but decreased level of levels of MCP-1, VCAM-1, and ICAM-1 in table 4 in control group. The results were contradictory, since lipid levels and endothelial function biomarkers are usually positively correlated, making the results unreliable.

4. The paper did not show how the sample size was calculated.

5. Lack of key information such as smoking, drinking, and physical activity. Differences in these factors have important effects on endothelial function and blood lipid.

Reviewer 2 Report

In this paper, authors try to evaluate the effect of black garlic consumption on markers of cardiovascular function in adult populations at risk of developing CVD, compared to the effects on healthy subjects. This is a very interesting and well-written paper, however I have some minor comments.

1) Was the sample size adequate for this conclusion?

2) It would be more clear for the reader if the authors find under each table a note of how the p-values were performed.

Overall, great work!

Reviewer 3 Report

Table 1. please chronologically

Table 2. illegible 

Table 3. illegible 

Table 4. illegible 

Abstract: to be changed

Title: to be changed

Numbering in the bibliography is not chronological

DOI numbers are missing

Table 1. please chronologically

Table 2. illegible 

Table 3. illegible 

Table 4. illegible 

Abstract: to be changed

Title: to be changed

Numbering in the bibliography is not chronological

DOI numbers are missing

Round 2

Reviewer 1 Report

Although the author emphasized some of my concerns, I would think that the quality of the article is still not suitable for the magazine